# RLTF: Reinforcement Learning from Unit Test Feedback

**Jiate Liu**[*]                                                    *jiateliu@tencent.com*
*Tencent*

**Yiqin Zhu**[*]                                                    *yiqinzhu@tencent.com*
*Tencent*

**Kaiwen Xiao**                                                    *loktarxiao@tencent.com*
*Tencent*

**Qiang Fu**                                                    *leonfu@tencent.com*
*Tencent*

**Xiao Han**                                                    *haroldhan@tencent.com*
*Tencent*

**Wei Yang**                                                    *willyang@tencent.com*
*Tencent*

**Deheng Ye**[†]                                                    *dericye@tencent.com*
*Tencent*

Reviewed on OpenReview: `https://openreview.net/forum?id=hjYmsV6nXZ&noteId=7gJrNDYNSs`

## Abstract

The goal of program synthesis, or code generation, is to generate executable code based on given descriptions. Recently, there has been an increasing number of studies employing reinforcement learning (RL) to improve the performance of large language models (LLMs) for code. However, some of the current representative RL methods have only used offline frameworks, limiting the exploration of new sample spaces. Additionally, the utilization of unit test signals is limited, not accounting for specific error locations within the code. To address these issues, we proposed RLTF, i.e., Reinforcement Learning from Unit Test Feedback, a novel online RL framework with unit test feedback of multi-granularity for refining code LLMs. Our approach generates data in real-time during training and simultaneously utilizes fine-grained feedback signals to guide the model towards producing higher-quality code. Extensive experiments show that RLTF achieves state-of-the-art performance on the APPS and the MBPP benchmarks. Our code is available at: `https://github.com/Zyq-scut/RLTF`.

## 1 Introduction

Program synthesis, or code generation, involves creating an executable program that solves a given problem. This research area has gained attention due to its potential to improve productivity and accessibility in programming. An AI model that can generate programs based on human requirements could significantly transform programming tools and workflows (Zan et al., 2023).

---

[*]Co-first authors.
[†]Corresponding author.

Recently, there has been a significant increase in the development of large language models (LLMs) built on Transformer (Vaswani et al., 2017) architectures, trained on large unlabelled code datasets. These formidable models possess the power to generate code without explicit programming instructions and have shown remarkable results in program synthesis. Codex (Chen et al., 2021) is a noteworthy example of an LLM with 12 billion parameters that can successfully solve over 70% of complex Python programming challenges created by humans. Moreover, these models have been proven to enhance productivity and coding effectiveness in real-world applications. Since Codex's emergence, many other LLMs tailored for the code domain have appeared, ranging in model size from millions to billions of parameters. AlphaCode (Li et al., 2022), for instance, aspires to address competitive-level programming challenges, while InCoder (Fried et al., 2022) enables code insertion at arbitrary junctures utilizing bidirectional contexts. Other acclaimed models include CodeT5 (Wang et al., 2021), CodeGen (Nijkamp et al., 2022), PaLM-Coder (Chowdhery et al., 2022), PanGu-Coder (Christopoulou et al., 2022), CodeGeex (Zheng et al., 2023), and SantaCoder (Allal et al., 2023). As the size of these LLMs increases, they demonstrate emergent competencies, including human-like programming prowess and debugging aptitude (Zhang et al., 2022; Saunders et al., 2022).

While LLMs have shown promising results in basic programming tasks, there is still progress to be made in tackling more challenging problems such as program competitions. Additionally, pretrained code models can be limited by their reliance on NLP's self-supervised masked language modeling (MLM) and may struggle with ensuring the syntactic and functional correctness of generated code. To improve code generation performance, very recently, reinforcement learning (RL) based algorithms for better utilizing code LLMs have been proposed. For example, CodeRL (Le et al., 2022) has explored the integration of reinforcement learning with unit test signals to fine-tune program synthesis models, and proposes a post-process method called "Critic Sampling" to further improve the performance of the model. PPOCoder (Shojaee et al., 2023) uses the Proximal Policy Optimization algorithm to improve CodeRL, although it does not achieve better results on the competition algorithm benchmark APPS, compared to CodeRL.

It is worth noting that some of existing RL-based methods (Le et al., 2022) are offline, meaning that they do not interact with the environment dynamically during training, but rather learn from pre-collected data. This can result in RL training quality being limited by the diversity of the dataset and hinder the effective exploration of the environment (Fujimoto et al., 2019; Kumar et al., 2019). Studies have also shown that offline RL can face issues such as distribution shift (Fujimoto et al., 2019; Schrittwieser et al., 2021), leading to unstable training and suboptimal policy performance. In contrast, online RL refers to the scenario where an agent can interact with the environment in real-time, selecting actions and observing their impact to obtain state and reward signals. Compared to offline RL, online RL training is often more stable, enables better exploration of the environment, and leads to the development of more optimal policies. On the other hand, while existing RL-based methods do employ the results of unit tests as feedback signals, their implementation is rather simple and coarse-grained. They assign the same reward to the entire episode (a complete code segment) based on the program's submission result (true, false, runtime error, or compiler error), without considering that a part of the code may be correct while another part contains bugs. As a result, these approaches fail to capture the nuances in identifying individual code components that contribute to the overall functionality and fixing bugs specific to certain parts of the code. Additionally, in contrast to domains such as healthcare, autonomous driving, and human-aligned in LLMs (Levine et al., 2020; Prudencio et al., 2023), where real-time interaction costs are high or data is difficult to obtain in real-time, the feedback from unit tests in the program synthesis task does not incur notable costs. This can potentially make the use of an online framework even more beneficial in this context.

To address the aforementioned issues, we introduce RLTF, a novel online framework with multi-granularity unit test feedback that enhances pretrained LLMs for program synthesis using reinforcement learning. First, we develop an online framework, which consists of two main parts: one for training the model and updating the parameters, and the other for generating programs using the latest model parameters and appending them to the online buffer for the subsequent model training. This framework enables the model to obtain real-time unit test feedback. Additionally, it provides a diverse set of samples, enabling the code LLM agent to explore the environment effectively while also promoting training stability. Second, we have analyzed the distribution of error types in programs and extracted more detailed information from unit test feedback for training, such as "fine-grained feedback" and "adaptive feedback". Fine-grained feedback categorizes

Table 1: RLTF and PPOCoder adopt an online framework, which allows the model to generate new samples in real-time, while CodeRL rely solely on offline frameworks. RLTF extracts more detailed information from unit test feedback for training, such as fine-grained feedback and adaptive feedback. Both RLTF and CodeRL use a post-processing technique called Critic Sampling, which was first introduced in (Le et al., 2022).

| Feature | CodeRL (Le et al., 2022) | PPOCoder (Shojaee et al., 2023) | RLTF |
|---|---|---|---|
| Framework | offline | online | online |
| Supervised Learning | ✓ | ✓ | ✓ |
| Coarse-grained Feedback | ✓ | ✓ | ✓ |
| Fine-grained Feedback | × | × | ✓ |
| Adapative Feedback | × | × | ✓ |
| Critic Sampling | ✓ | × | ✓ |

errors based on their reported information and location, penalizing the specific erroneous parts of the code accordingly. Adaptive feedback provides varying rewards to programs based on the ratio of test cases they pass. More detailed information can be found in Sec. 3.3.2.

The main contributions of our paper are as follows:

- We propose RLTF, a novel online framework for the program synthesis task. By generating new samples in real-time during the training process and leveraging unit test results as feedback signals, it improves overall model performance. This approach enables models to learn the specifics of code errors and enhance their performance accordingly.

- Built upon this framework, we develop multi-granularity feedback that is automatically extracted from unit test. To expand, we introduce coarse-grained and fine-grained feedbacks applicable to programs with errors, aimed at punishing the specific segments of code where the errors appear. For programs that do not pass all test cases, we propose an adaptive feedback mechanism that assigns varying penalties based on the ratio of passed test cases.

- For the program synthesis task, our method achieves state-of-the-art results on the APPS and MBPP benchmarks, and a detailed ablation study demonstrates the effectiveness of our approach. Additionally, we perform tests on different LLMs (e.g., CodeT5, CodeGen), illustrating the robustness of our method and its applicability to different base models.

## 2 Related Work

### 2.1 Pretrained LLMs for Program Synthesis

Recent research has delved into leveraging pretrained large language models (LLMs) from the natural language processing (NLP) field to automate program synthesis tasks, using vast-scale code corpus data mined from open-source repositories. Notably, there are several prominent examples of such pretrained models including the encoder-only CodeBERT (Feng et al., 2020), decoder-only CodeGPT (Lu et al., 2021), CodeGen (Nijkamp et al., 2022), PaLM-Coder (Chowdhery et al., 2022), PanGu-Coder (Christopoulou et al., 2022), CodeGeex (Zheng et al., 2023), and SantaCoder (Allal et al., 2023), as well as encoder-decoder transformer architectures like PLABRT (Ahmad et al., 2021) and CodeT5 (Wang et al., 2021). These pretrained probabilistic language (PL) models are already capable of generating code that appears visually impressive and well-structured. However, these models still face challenges in guaranteeing the syntactic and functional correctness of the generated codes (Chen et al., 2021; Hendrycks et al., 2021; Shojaee et al., 2023).

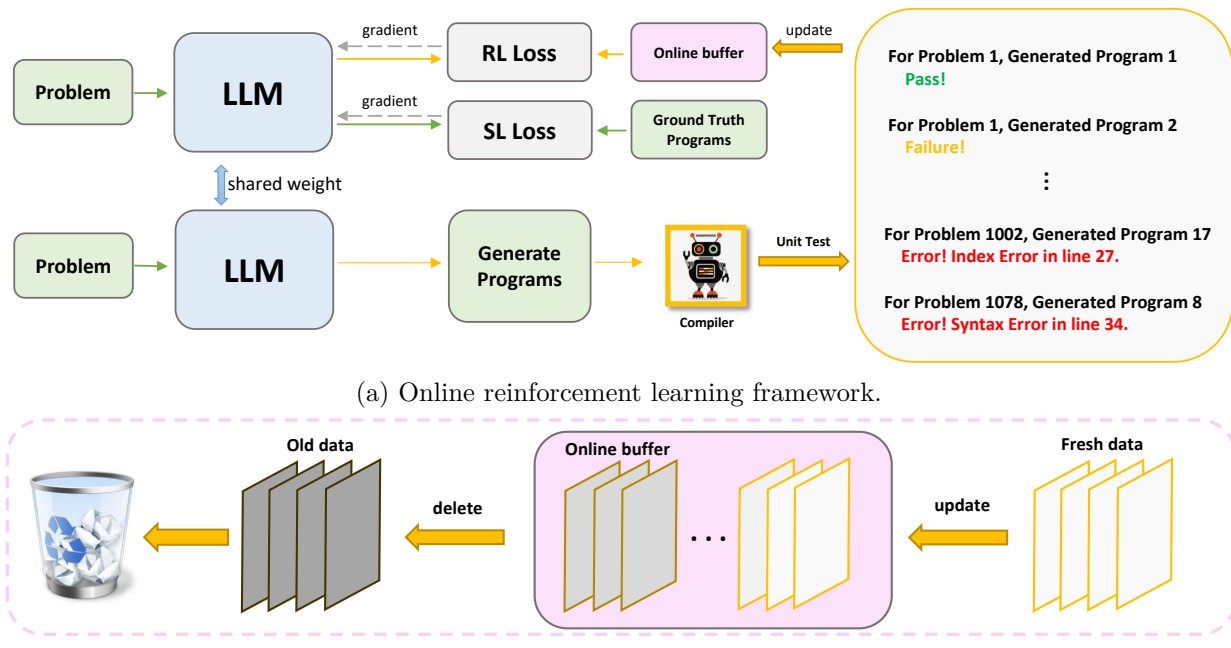

(a) Online reinforcement learning framework.

(b) Online buffer workflow.

Figure 1: The overall framework of the proposed RLTF. (a) Online reinforcement learning framework: Two LLMs with shared weight are utilized in the training process. One generates the target program and interacts with the compiler to produce a training data pair, which is then stored in the online buffer. The other one utilizes ground truth data and online buffer data to calculate loss and update the model weights through gradient feedback. (b) Online buffer workflow: Online buffers are utilized to retain the newly generated data for RL training. As fresh data is received, old data is deleted. Here SL refers to supervised learning and RL refers to reinforcement learning.

## 2.2 Reinforcement Learning for Program Synthesis

Reinforcement Learning (RL) is a method of learning the optimal policy by obtaining reward signals from the real environment (Sutton et al., 1998). In recent years, RL has shown promise in optimizing non-differentiable metrics for sequence generation tasks, such as improving BLEU and ROUGE scores in summarization and translation models respectively (Husain et al., 2019; Ren et al., 2020). Unlike text generation, program synthesis requires both syntactic and functional accuracy since the generated code must pass compilation and unit tests (Hendrycks et al., 2021; Chen et al., 2021). Recently, execution-guided approaches and RL-based fine-tuning mechanisms have been employed to enhance the quality of generated codes. For instance, CodeRL (Le et al., 2022) investigates the fusion of reinforcement learning and unit test signals to enhance program synthesis models, introducing a post-processing technique named Critic Sampling that further boosts model performance. PPOCoder (Shojaee et al., 2023) employs the Proximal Policy Optimization algorithm from reinforcement learning to refine CodeRL. However, existing RL-based methods still face several limitations. First, some of the existing RL methods have only used offline frameworks, which constrain their ability to explore new sample spaces. Moreover, the current techniques employing unit test signals are relatively simple, neglecting to consider specific error information and locations within the code. Our RLTF method adopts an online framework and incorporates multi-grained feedback for model training. Overall, the differences between RLTF and existing approaches such as CodeRL and PPOCoder are summarized in Table 1.

## 3   Approach

### 3.1   Task Definition

Program synthesis is an automated process that generates computer code $W$ based on a high-level description of desired behavior $D$. This process aims to increase the likelihood of generating a code that can solve a given problem, expressed as a conditional probability $P(W|D)$.

The task can be expressed as a sequence-to-sequence problem, and the objective is to maximize the conditional probability of generating the correct output, given the input and the LLM model parameters:

$$\max P(W|D,\theta) = \max \prod_{t=1}^{T} p(w_t|D,\theta,w_{1:t-1}) \tag{1}$$

where $\theta$ is the parameters of the LLM model. The LLM model is used to capture the conditional distribution, and it is trained using a set of input-output pairs. During training, the parameters $\theta$ are adjusted to maximize the likelihood of generating the correct output for a given input.

### 3.2   Online Reinforcement Learning Framework

While supervised learning can yield valuable outcomes, the vast search space involved in program synthesis can make it challenging to acquire a comprehensive dataset. As such, we propose an online reinforcement learning framework, which continuously improves the code ability of LLM through online generated training samples and fine-grained feedback provided by the compiler, to circumvent this predicament. Specifically, our framework incorporates an online buffer during the training process, which is crucial in improving the model's effectiveness. The buffer maintains a dynamic flow of data pairs that are utilized in reinforcement learning training in real-time. Each data pair comprises a problem description $D$, the most recent model-generated code $\hat{W}$, and feedback from the compiler $FB(\hat{W})$. This approach allows the model to adapt to changes in the data distribution and facilitate more efficient learning. The present framework incorporates the actor and critic in training, similar to the methodology employed in CodeRL (Le et al., 2022) and PPOCoder (Shojaee et al., 2023).

The overall framework is shown in Fig. 1. The training process involves two LLMs with shared weights. The first LLM updates model parameters using data obtained from buffer for training, while the second LLM collaborates with the compiler to dynamically generate code, evaluate its soundness, generate data pairs, and continuously refresh buffer in real-time. The incorporation of a continuously updated buffer ensures the promptness of the data used for training, thereby enabling the model to explore a wider sample space comprehensively.

### 3.3   Reinforcement Learning from Unit Test Feedback

We propose RLTF, a method to improve the quality of synthesized codes by exploiting unit test feedback to explore the target space. Following the reinforcement algorithm Sutton et al. (1998)Sutton & Barto (2018), we construct reinforcement learning loss with the explored code $\hat{W}$ as follows:

$$L_{rl} = -R(\hat{W}) \log P(\hat{W}|D,\theta) = -R(\hat{W}) \sum_{t=S}^{E} \log p(\hat{w}_t|D,\theta,\hat{w}_{1:t-1}) \tag{2}$$

where $R(*)$ is the reward coefficient, $S$ and $E$ represent the start and end positions of the penalized code snippets. Their values are determined by different feedback categories.

#### 3.3.1   Feedback Category

Unit test feedback $FB(\hat{W})$ can be obtained from the program compiler and can be categorized into three main groups, namely *Error*, *Failure*, and *Pass*.

Table 2: Definition of sub-error types in Python. We divide different sub-errors into three categories based on their characteristics: $U_{global}$, $U_{line}$, and $U_{ignore}$.

| Sub-error | Description | Category |
|---|---|---|
| Syntax Error | Code contains syntax errors that cause the compilation to fail | $U_{line}$ or $U_{global}$ |
| Index Error | Index operation is out of bounds | $U_{line}$ |
| Type Error | An operation or function was applied to an object of an inappropriate type | $U_{line}$ |
| Value Error | An operation or function received an argument with the correct type but with an inappropriate value | $U_{line}$ |
| EOF Error | The input() function encountered an end-of-file condition (EOF) without reading any data | $U_{line}$ |
| Timeout Error | Code execution time exceeds time limit | $U_{global}$ |
| Name Error | A local or global name is not defined | $U_{line}$ |
| Key Error | A mapping (dictionary) key is not found in the set of existing keys | $U_{line}$ |
| Import Error | The imported package is not found | $U_{line}$ |
| ZeroDivision Error | The second argument of a division or modulo operation is zero | $U_{line}$ |
| Recursion Error | Code execution recursive operation exceeds the maximum limit | $U_{global}$ |
| Triple-quoted Error | Triple quotes are incomplete | $U_{ignore}$ |
| Indentation Error | Wrong indentation format | $U_{ignore}$ |
| Else | - | $U_{line}$ |

*Error* refers to a situation where the generated code fails to compile or results in errors such as exceptions that prevent the program from functioning normally. This category can be further subdivided into syntax errors, index errors, type errors, among others.

*Failure* indicates that the generated code can be compiled or run, but it fails to meet the expected functional or performance requirements. This situation requires additional improvement or optimization of the program. Code belonging to this category fails at least one unit test.

*Pass* implies that the generated code can be compiled and run successfully and meets the expected functional and performance requirements. This category indicates that the program has passed all unit tests and can be used without further modification.

### 3.3.2 Multi-granularity Feedback

**Coarse-grained Feedback.** The coarse-grained feedback serves as an incentive mechanism for the language model to enhance the probability of producing accurate code while reducing the likelihood of generating erroneous code. We use the same setting as CodeRL (Le et al., 2022) and PPOCoder (Shojaee et al., 2023):

$$R_{coarse}(\hat{W}) = \begin{cases} 1.0, & FB(\hat{W}) \text{ is pass} \\ -0.3, & FB(\hat{W}) \text{ is failure} \\ -0.6, & FB(\hat{W}) \text{ is error except syntax error} \\ -1.0, & FB(\hat{W}) \text{ is syntax error} \end{cases}, S_{coarse} = 0, E_{coarse} = T \qquad (3)$$

The value of $R_{coarse}$ is determined by complier feedback $FB(\hat{W})$. The start position $S_{coarse}$ and end position $E_{coarse}$ are always 0 and $T$.

**Fine-grained Feedback.** The coarse-grained feedback serves to inform the model that it has made an error, but does not indicate the specific location of the error. In contrast, the purpose of the fine-grained feedback is to provide the model with information regarding the specific reasons for its errors, with the goal of reducing the likelihood of future errors. The reasons for code errors can vary widely, sometimes stemming from specific errors in the code (such as the use of undefined variables), and at other times from issues with the overall logic of the code (such as recursion errors in Python). For the former, we aim for the model to address the error on the current line, while for the latter, we hope the model can re-evaluate and revise the

code. To this end, we have developed a set of rewards for different error subtypes, which can be found in Table 2.

Typically, we are able to identify the erroneous code snippets from compiler prompts. We divide the error codes into three categories ($U_{global}, U_{line}, U_{ignore}$) based on their specific subtypes. The $U_{global}$ category means that penalties will be applied to the entire code generated, usually because of a logic issue that affects the entire code segment, and cannot be exactly identified on a specific line of code. The $U_{line}$ category applies penalties to the specific line of code that caused the error because fixing this line of code will prevent that error from recurring. The $U_{ignore}$ category temporarily avoids punishment because the reasons for the error are numerous and cannot be accurately located. In cases of Syntax Error, if it is caused by unfinished code due to the maximum token length limit, it is classified as $U_{global}$, otherwise, it is $U_{line}$.

$$R_{fine}(\hat{W}) = \begin{cases} 0.0, \text{if } \hat{W} \in U_{ignore} \\ -0.3, \text{else} \end{cases}, S_{fine} = \begin{cases} t_{line\_start}, \text{if } \hat{W} \in U_{line} \\ 0, \text{else} \end{cases}, E_{fine} = \begin{cases} t_{line\_end}, \text{if } \hat{W} \in U_{line} \\ T, \text{else} \end{cases} \tag{4}$$

We set $R_{fine}(\hat{W}) = 0.0$ for $U_{ignore}$, otherwise $-0.3$. $t_{line\_start}$ and $t_{line\_end}$ are the start and end position of the line that complier feedback.

**Adaptive Feedback.** Adaptive feedback is designed to motivate the model to generate code that can successfully pass a maximum number of test samples. To attain this objective, the reward value of adaptive feedback is directly proportional to the number of test samples that the generated code can pass. The specific configuration of the reward is as follows:

$$R_{adaptive}(\hat{W}) = -0.3 + 1.3 * \frac{N_{pass}}{N_{pass} + N_{fail}}, S_{adaptive} = 0, E_{adaptive} = T \tag{5}$$

The value of $R_{adaptive}$ is positively correlated with the pass rate of unit tests. The start and end position are always 0 and T.

### 3.4 Optimization Objective

In order to further enhance the performance of the pre-trained LLM via RLTF, the following optimization objectives can be used for fine-tuning:

$$L_{total} = L_{sl} + L_{coarse} + L_{fine} + L_{adaptive} \tag{6}$$

where $L_{sl}$ is supervised learning loss which makes the training process stable. We adopt cross-entropy loss as $L_{sl}$ as follows:

$$L_{sl} = -\log P(W|D, \theta) = -\sum_{t=1}^{T} \log p(w_t|D, \theta, w_{1:t-1}) \tag{7}$$

And $L_{coarse}, L_{fine}, L_{adaptive}$ are variants of the reinforcement learning loss shown in Eq.2, representing three types of feedback granularity, which are as follows:

$$L_{coarse} = -(R_{coarse}(\hat{W}) - R_{coarse}(\hat{W}_{baseline})) \sum_{t=S_{coarse}}^{E_{coarse}} \log p(\hat{w}_t | D, \theta, \hat{w}_{1:t-1}) \qquad (8)$$

$$L_{fine} = -\alpha R_{fine}(\hat{W}) \sum_{t=S_{fine}}^{E_{fine}} \log p(\hat{w}_t | D, \theta, \hat{w}_{1:t-1}) \qquad (9)$$

$$L_{adaptive} = -(R_{adaptive}(\hat{W}) - R_{adaptive}(\hat{W}_{baseline})) \sum_{t=S_{adaptive}}^{E_{adaptive}} \log p(\hat{w}_t | D, \theta, \hat{w}_{1:t-1}) \qquad (10)$$

To address the amplitude fluctuation in fine-grained feedback caused by variable values of $S$ and $E$, we use an adaptive weight $\alpha$ that balances the equation. This weight is calculated as $\alpha = \frac{T}{E_{fine} - S_{fine}}$, making the fine-grained feedback equivalent to coarse-grained feedback and adaptive feedback. Here $\hat{W}_{baseline}$ is the baseline dynamically updated during the training process, which represents the historical optimal code generated by the model under the natural language description $D$. This item indicates that we want the model to continuously outperform its own historical version. For fine-grained feedback, its baseline is hard to define, so we do not add this item to its loss function.

## 4 Experiments

Our baselines are the state-of-the-art works on incorporating RL with code LLMs (PPOCoder (Shojaee et al., 2023) and CodeRL (Le et al., 2022)), and we use the same benchmarks and setting as theirs for evaluation.

### 4.1 Benchmarks

**APPS Benchmark.** We first evaluate using the challenging APPS (Automated Programming Progress Standard) program synthesis benchmark presented by (Hendrycks et al., 2021), as it features a diverse range of coding problems from various coding websites, presenting differing levels of difficulty. The benchmark consists of a total of 10,000 coding problems, with an equal train-test split. On average, each problem is accompanied by 23.2 accurate Python programs and 21.2 unit tests. The mean length of each problem is 293.2 words, whereas the average program length is 18.0 lines. The dataset is classified into three difficulty levels: Introductory (3,639; train/test = 2,639/1,000), Interview (5,000; train/test = 2,000/3,000), and Competition (1,361; train/test = 361/1,000). Typically, each sample has 20 unit tests to assess the functional correctness of the programs. We adhere to the same preprocessing steps as those in (Hendrycks et al., 2021) for generating input sequences based on problem descriptions.

Note that, during the implementation of our online framework, we encountered an issue in the APPS open-source code, specifically in the unit test portion. If the generated code raised a "segmentation fault" error, it could not be bypassed using "try" and "except", resulting in the main program getting stuck and disrupting the sample generation process. To address this issue, we modified all unit test code procedures to be executed through subprocess calls, ensuring the main program does not get stuck and allowing the online sample generation process to run smoothly. We believe it necessary to expose this to the community for reproducibility considerations.

For the APPS benchmark, we employ the RLTF framework to fine-tune the pretrained CodeT5 model. We utilized a machine equipped with 8 NVIDIA V100 GPUs, each with 32GB of memory, for training purposes. Each GPU carried a batch size of 32, and a learning rate of 2e-6 was employed. The training process took approximately 24 hours. Concurrently, three additional machines with similar 8-card V100 GPU configurations were used to generate the latest samples. We updated the online buffer every 50 steps. Following the same approach as CodeRL, half of the steps were for SL training, while the other half focused on RL training. The length of the online buffer was set to 6400. After generating 6400 samples using the initial model, the training process officially commenced, and the online buffer was updated as a queue.

Table 3: Quantitative evaluation on APPS benchmark.

| Method | Size | CS | pass@1 | | | | pass@5 | | | | pass@1000 | | | |
|---|---|---|---|---|---|---|---|---|---|---|---|---|---|---|
| | | | Intro | Inter | Comp | all | Intro | Inter | Comp | all | Intro | Inter | Comp | all |
| Codex | 12B | w/o | 4.14 | 0.14 | 0.02 | 0.92 | 9.65 | 0.51 | 0.09 | 2.25 | 25.02 | 3.70 | 3.23 | 7.87 |
| AlphaCode | 1B | w/o | - | - | - | - | - | - | - | - | 17.67 | 5.24 | 7.06 | 8.09 |
| GPT3 | 175B | w/o | 0.20 | 0.03 | 0.00 | 0.06 | - | - | - | - | - | - | - | - |
| GPT2 | 0.1B | w/o | 1.00 | 0.33 | 0.00 | 0.40 | 2.70 | 0.73 | 0.00 | 1.02 | - | - | - | - |
| GPT2 | 1.5B | w/o | 1.30 | 0.70 | 0.00 | 0.68 | 3.60 | 1.03 | 0.00 | 1.58 | 27.90 | 9.83 | 11.40 | 13.76 |
| GPT-Neo | 2.7B | w/o | 3.90 | 0.57 | 0.00 | 1.12 | 5.50 | 0.80 | 0.00 | 1.58 | 27.90 | 9.83 | 11.40 | 13.76 |
| CodeRL | 770M | w/o | 4.00 | 0.78 | 0.15 | 1.30 | 9.83 | 2.03 | 0.69 | 3.32 | 35.30 | 13.33 | 13.60 | 17.78 |
| PPOCoder(original) | 770M | w/o | 5.20 | 1.00 | 0.50 | 1.74 | 9.10 | 2.50 | 1.20 | 3.56 | 35.20 | 13.35 | 13.90 | 17.77 |
| PPOCoder(scale) | 770M | w/o | 4.06 | 0.79 | 0.15 | 1.32 | 9.97 | 2.06 | 0.70 | 3.37 | 35.42 | 13.37 | 13.65 | 17.84 |
| RLTF | 770M | w/o | 4.16 | 0.97 | 0.20 | 1.45 | 10.12 | 2.65 | 0.82 | 3.78 | 38.30 | 15.13 | 15.90 | 19.92 |
| CodeRL | 770M | w | 7.08 | 1.86 | 0.75 | 2.69 | 16.37 | 4.95 | 2.84 | 6.81 | **40.00** | **15.67** | **17.90** | **20.98** |
| RLTF | 770M | w | **8.40** | **2.28** | **1.10** | **3.27** | **18.60** | **5.57** | **3.70** | **7.80** | 39.70 | 15.03 | 16.80 | 20.32 |

During testing, we employed Nucleus sampling with a top value of 0.95 and set the temperature parameter to 0.6.

**MBPP Benchmark.** To further evaluate our framework, we also employ an additional, smaller, and simpler Python program synthesis dataset called MBPP (Mostly Basic Programming Problems), introduced by (Austin et al., 2021). The dataset consists of 974 instances, with 374 instances for training, 90 instances for validation, and 500 instances for testing, while reserving 10 instances for few-shot learning evaluations. The problems are predominantly brief, often featuring only a single sentence of natural language descriptions. Each problem is accompanied by one correct solution (averaging 6.8 lines of code) and three unit tests, which are assert statements to verify functional correctness. In contrast to APPS, MBPP's unit tests are not hidden and are explicitly integrated into the source sequences for program synthesis models. Although this occasionally encourages models to overfit the assert statements through hard-coding an if-expression, we adhere to the same source sequence construction as previous work to ensure a fair comparison with baselines.

Specifically, we use the same prompt format as (Austin et al., 2021) to prepare the input sequence, which is composed of problem descriptions + "Your code should satisfy these tests:" + three assert statements. We experiment with the MBPP dataset in a zero-shot setting, as the same as CodeRL. While generating samples, we also used Nucleus sampling with a top value of 0.95 and set the temperature parameter to 1.2.

## 4.2 Quantitative Evaluation on APPS

For a fair comparison, we employ the CodeT5 770M as the base model, similarly to CodeRL (Le et al., 2022), PPOCoder(Shojaee et al., 2023). Table 3 presents the overall results of our RLTF approach based on the CodeT5 model on the APPS benchmark. We compared our method with larger models such as Codex (Chen et al., 2021), AlphaCode (Li et al., 2022), GPT2 (Radford et al., 2019), GPT3 (Brown et al., 2020), GPT-Neo (Black et al., 2021), and other CodeT5-based methods like CodeRL (Le et al., 2022) and PPOCoder (Shojaee et al., 2023). Note that by default, results of pretrained LMs (except for Codex and GPT3) are from models fine-tuned on APPS using the standard loss $L_{ce}$ only, and results of CodeRL, PPOCoder, RLTF are from models fine-tuned on APPS using the both loss $L_{ce}$ and $L_{rl}$. The results indicate that methods based on CodeT5 outperform those using other larger-parameter base models such as GPTs. Furthermore, employing the RLTF approach leads to additional performance improvements, surpassing other CodeT5-based methods such as CodeRL and PPOCoder, thereby achieving a new state-of-the-art (SOTA) in the field. It is important to note that we utilized the official open-source CodeRL model and applied nucleus sampling with a temperature of 0.6. We noticed that our pass@1 score was lower than the result reported in their paper, while our pass@5 score was higher. Similar results were also found by others and discussed at https://github.com/salesforce/CodeRL/issues/30. Regarding PPOCoder, the results reported in their paper also differ from those in the CodeRL paper. Since the authors did not open-source the APPS benchmark code and models, we are uncertain about their exact evaluation process. Therefore, we used the results obtained from the open-source CodeRL model as a standard and employed proportional scaling to

Table 4: Ablation studies: Impact of framework.

| Framework | pass@1 | | | | pass@5 | | | | pass@10 | | | |
|---|---|---|---|---|---|---|---|---|---|---|---|---|
| | Intro | Inter | Comp | all | Intro | Inter | Comp | all | Intro | Inter | Comp | all |
| Offline | 3.67 | 0.84 | 0.16 | 1.29 | 9.25 | 2.40 | 0.74 | 3.43 | 12.41 | 3.36 | 1.33 | 4.76 |
| Offline+RLTF | 3.71 | 0.93 | 0.19 | 1.34 | 9.28 | 2.51 | **0.85** | 3.53 | 12.60 | 3.50 | **1.50** | 4.92 |
| Online | 3.94 | 0.92 | 0.17 | 1.37 | 9.68 | 2.43 | 0.75 | 3.50 | 13.01 | 3.41 | 1.35 | 4.92 |
| Online+RLTF | **4.16** | **0.97** | **0.20** | **1.45** | **10.12** | **2.65** | 0.82 | **3.78** | **13.59** | **3.67** | 1.45 | **5.21** |

derive the final results. To provide readers with a clear understanding of the result sources, we concurrently included the results for PPOCoder(original) and PPOCoder(scale) in Table 3. Here, PPOCoder(original) represents the results from the original paper; however, since we did not directly compare against it, we have shaded it in gray. PPOCoder(scale), on the other hand, denotes the scaled results. This ensures a fair comparison between the different methods, while acknowledging the discrepancies in the reported values. Even without selecting the scaled results, our performance is better than the results reported in the original paper, except for pass@1.

Additionally, we compare the performance of our model with the post-processing approach Critic Sampling, first proposed in (Le et al., 2022). Due to the unavailability of the open-source Critic Sampling code from CodeRL, we reproduced the Critic Sampling based on the details provided in the original paper. We employed both refine and repair and set $M$, the number of top candidates selected from program samples that fail example unit tests to 1. We also limited the maximum number of repair and refine iterations to 1. It is worth noting that we found a crucial parameter $N$ not explicitly mentioned in the original paper: the total number of program samples generated during each refine/repair step. When measuring $pass@1$, we set $N = 5$; for $pass@5$, we set $N = 20$; and for $pass@1000$, we set $N = 1000$. Our results show that Critic Sampling achieves better performance when $k = 1$ or 5 in $pass@k$. However, when $k$ is larger, the improvement in performance is relatively small, which differs from the results presented in the original paper. Overall, we think Critic Sampling is a post-processing method that focuses on how to better use the model rather than how to better train the model. Therefore, comparing our approach with results without Critic Sampling provides a fairer evaluation.

### 4.3 Ablation Studies

We conduct extensive ablations to understand the effectiveness of the techniques we develop.

**Impact of Framework.** Table 4 presents the ablation study comparing different training frameworks (online or offline) and the use of the RLTF method. All the results have utilized the combined loss $L_{sl}$ and $L_{coarse}$. The results show that the combination of the online framework and RLTF yields the best performance, while the offline framework without RLTF produces the poorest outcome. Applying either the online framework or RLTF method individually leads to a certain degree of improvement, validating the effectiveness of both approaches. Interestingly, we observe that the performance boost brought by the RLTF method is more substantial within the online framework. This can be attributed to the real-time generation of samples in the online setting, which helps the model identify more potential errors in programs and thus facilitates better training.

**Impact of Feedback.** Table 5 presents the ablation study comparing different feedback combinations during reinforcement learning (RL) training. The results show that using only supervised learning yields the poorest performance. In contrast, employing reinforcement learning and progressively incorporating the rewards mentioned in Section 3 leads to increasingly better model performance. The best results are obtained when using a combination of coarse-grained, fine-grained, and adaptive feedback, verifying the effectiveness of these different reward designs. Additionally, we observe that the inclusion of the fine-grained feedback contributes the most significant performance boost among the rewards evaluated. We also evaluate the results with all feedback except the coarse one: $pass@1$ 1.38%, $pass@5$ 3.58%, $pass@10$ 5.05%. The purpose of the coarse feedback is to inform the model about the overall test results of the code (true, false, runtime

Table 5: Ablation: impact of different combinations of feedback.

| SL | Coarse | Fine | Adaptive | pass@1 | pass@5 | pass@10 | pass@100 | pass@1000 |
|----|--------|------|----------|--------|--------|---------|----------|-----------|
| √ | - | - | - | 1.30 | 3.39 | 4.68 | 10.51 | 17.80 |
| √ | √ | - | - | 1.37 | 3.50 | 4.92 | 10.63 | 18.31 |
| √ | √ | √ | - | 1.41 | 3.67 | 5.10 | 11.08 | 19.32 |
| √ | √ | √ | √ | **1.45** | **3.78** | **5.21** | **11.23** | **19.92** |

Table 6: Ablation: impact of the fine-grained feedback $R_{fine}(\hat{W})$.

| $R_{fine}(\hat{W})$ | pass@1 | pass@5 | pass@10 |
|----------|--------|--------|---------|
| 0.0 | 1.31 | 3.42 | 4.72 |
| -0.1 | 1.39 | 3.59 | 4.99 |
| -0.2 | 1.38 | 3.67 | 5.10 |
| -0.3 | 1.40 | 3.62 | 5.04 |
| -0.4 | 1.39 | 3.68 | 5.06 |
| -0.5 | 1.36 | 3.57 | 5.00 |

Table 7: Ablation: impact of the temperature during training.

| Temperature | pass@1 | pass@5 | pass@10 |
|-------------|--------|--------|---------|
| 0.2 | 1.34 | 3.56 | 4.99 |
| 0.6 | 1.37 | 3.60 | 5.06 |
| 1.0 | 1.45 | 3.78 | 5.21 |

error, compile error). Therefore, we think it is also quite important. As can be seen, removing the coarse feedback leads to a decline in the performance metrics. It is also worth noting that the performance without coarse feedback is better than the performance with only coarse feedback.

**Impact of $R_{fine}(\hat{W})$.** Table 6 illustrates the impact of different $R_{fine}(\hat{W})$ for fine-grained feedback on model training. We can observe that various values of $R_{fine}(\hat{W})$ result in improved model performance, with each offering better outcomes than those obtained without any penalty.

**Impact of Temperature.** Table 7 demonstrates the effect of varying the temperature during the training process. A higher temperature leads to the generation of more diverse samples, while a lower temperature results in more conservative samples. We observe that the model performs better when the temperature is set to 1, and comparatively worse when set to 0.2. This suggests that generating more diverse samples during the training process helps the model explore a broader range of sample space, facilitating RL training.

**Impact of Language Model.** To demonstrate the robustness of the RLTF approach, we conduct experiments using another base model, CodeGen 2.7B, in addition to CodeT5. As shown in Table 8, the results indicate that applying the RLTF method on CodeGen 2.7B also yields impressive performance, resulting in an almost 1% improvement in pass@10. Notably, we find that the larger the base model, the greater the performance boost provided by RLTF. This suggests that the RLTF approach can effectively unleash the potential of different base models to generate better codes, with the impact being more pronounced when the base model size is larger.

## 4.4 Quantitative Evaluation on MBPP

To further demonstrate the effectiveness of RLTF, we evaluate the zero-shot performance of the CodeT5 model trained with RLTF on the APPS benchmark when tested on the MBPP (Mostly Basic Python Problems) benchmark, as shown in Table 9. For this evaluation, the methods based on the CodeT5 model, including CodeRL, PPOCoder, and RLTF, were assessed in a zero-shot setting. For the GPT-based models, they were pretrained on 2.93 billion web documents using standard language modeling objective, and fine-tuned on the training set of the MBPP dataset. The results presented in the table were obtained from Figure 3 of the original paper (Austin et al., 2021). The results revealed that RLTF outperformed different sizes of GPT models on the MBPP benchmark and attained a new state-of-the-art performance among the CodeT5-based methods.

Table 8: Ablation: different large language models as the backbone.

| Language Model | RLTF | pass@1 | pass@5 | pass@10 | pass@100 | pass@1000 |
|---|---|---|---|---|---|---|
| CodeT5 770M | w/o | 1.30 | 3.39 | 4.68 | 10.51 | 17.80 |
| CodeT5 770M | w | **1.45** | **3.78** | **5.21** | **11.23** | **19.92** |
| CodeGen 2.7B | w/o | 1.64 | 4.23 | 5.79 | 12.48 | 21.44 |
| CodeGen 2.7B | w | **2.04** | **5.04** | **6.80** | **14.17** | **23.96** |

Table 9: Quantitative evaluation on the MBPP benchmark (zero-shot).

| Method | Size | state | pass@1 | pass@80 |
|---|---|---|---|---|
| GPT | 224M | fine-tuned | - | 7.2 |
| GPT | 442M | fine-tuned | - | 12.6 |
| GPT | 1B | fine-tuned | - | 22.4 |
| GPT | 4B | fine-tuned | - | 33.0 |
| GPT | 8B | fine-tuned | - | 40.6 |
| GPT | 68B | fine-tuned | - | 53.6 |
| GPT | 137B | fine-tuned | - | 61.4 |
| CodeT5 + CodeRL | 770M | zero-shot | 25.7 | 68.1 |
| CodeT5 + PPOCoder | 770M | zero-shot | 26.1 | 68.2 |
| CodeT5 + RLTF | 770M | zero-shot | **30.4** | **71.3** |

## 4.5 Qualitative Analysis by Unit Test Outcomes

Figure 2 presents the qualitative analysis by unit test outcomes before and after applying the RLTF method on the CodeGen model. For the 5000 problems in the APPS benchmark, we utilize Nucleus sampling to generate an average of 20 solutions per problem and recorded their results. Figure 2a depicts the percentages of different unit test results, as defined in Eq.3, including error, failure, and pass. We can observe that the RLTF method can reduce the proportion of programs resulting in errors and increase the proportion of programs that pass, especially for problems with introductory difficulty levels. The observed increase in failure rate stems from the fixing of error codes, resulting in either pass or fail outcomes. This illustrate that RLTF is more effective in addressing runtime and compiler errors compared to semantic errors, which remain more challenging. Figure 2b illustrates the percentages of different sub-errors among erroneous results before and after applying the RLTF method. Most errors show a decline in proportion after using the RLTF method, particularly syntax errors. It is also noteworthy that the proportion of timeout errors exhibits a minor increase, which can be attributed to the correction of other grammar-related errors resulting in timeout errors.

## 5 Conclusions and Future Work

In this paper, we have proposed RLTF (Reinforcement Learning from unit Test Feedback), a novel online RL framework with unit test feedback of multi-granularity, for refining large language models on program synthesis tasks. Compared with existing works, our approach generates data on-the-fly during training and simultaneously utilizes finer granularity in feedback signals to guide the model towards producing higher-quality code. Extensive experiments demonstrate that RLTF surpasses existing RL methods for programming and can be applied to various LLMs, including CodeT5 and CodeGen. Furthermore, it achieves state-of-the-art performance on widely-used benchmarks including APPS and MBPP.

In the future, there are several directions to further improve RLTF. For instance, the input-output examples in the existing benchmarks may not be sufficiently diverse, and the programs generated using hidden input-output examples may not be the correct final code versions. Such limitations can restrict the performance of RLTF. Therefore, using LLMs to create a more diverse and accurate set of input-output examples is a potential research direction worth exploring. Additionally, whether finer-grained feedback signals, such as those from static code analyzers, can further enhance RLTF's performance, is another possible avenue to pursue. Also, note that transferability is an importance issue. The manual categorization of sub-error

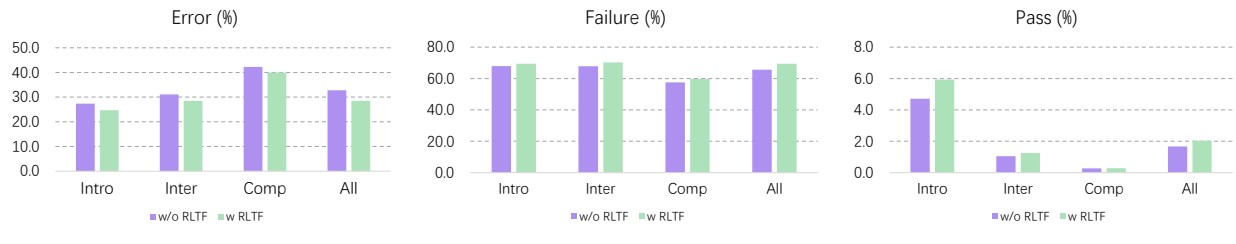

(a) Percentages of unit test results.

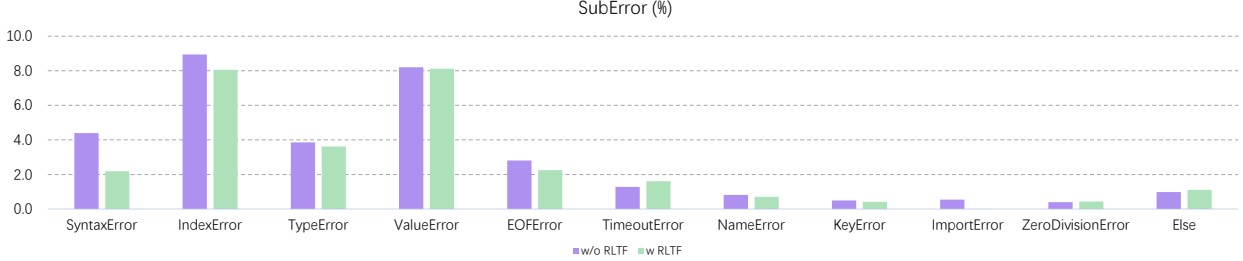

(b) Percentages of different sub-errors.

Figure 2: Qualitative analysis by unit test outcomes on CodeGen.

types we employed makes it challenging to transfer RLTF to other programming languages, which should be considered as another limitation of this work. As a next step, we believe the implementation of automated categorization will significantly enhance the applicability of RLTF in a broader context.

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
