# OpenReview forum: "RLTF: Reinforcement Learning from Unit Test Feedback"
_TMLR — Accepted by TMLR_

### Review · Reviewer_fP7d · 2023-08-09

**Summary Of Contributions:**

This paper introduces an online RL approach for improving code generating models based on feedback from unit test executions. It leverages both a coarse (execution level, based on whether the code compiles, passes tests) and fine-grained feedback structure (for code that raises exceptions), as well as a loss term based on test passing rate. Training CodeT5 large with this setup shows promising results on a benchmark of challenging programming problems.

**Audience:**

Yes

**Claims And Evidence:**

Yes

**Requested Changes:**

In line with the main weakness stated above, the work should evaluate the impact of its many design decisions. Specifically:

- Tab. 2/P6: involves a manual categorization of error types for just a subset of the errors in the Python programming language. This both means that porting this approach to new languages will be time-consuming and that it may not yield the observed improvements in systems with other error types. While those may be rare in interview problems, they abound in practice. The work should discuss this and note it as a limitation. Furthermore, the reason for having a separate U_ignore from U_global/U_whole is unconvincing. The motivation for this category is that "the reasons for the error are numerous and cannot be accurately located". That sounds very much like the definition of a global error. Please provide results where this issue type is grouped under global.
- The work uses 0.3 as a constant in most of its R_* loss terms. Please provide both (a) an analysis where each occurrence of this constant (and others in Eq. 3) is instead tuned on a validation set (a small grid search over permutations suffices) and the optimal values are applied to the test set, as well as (b) an ablation of each coefficient separately other than R_fine in the style of Tab. 6. The goal is to show how sensitive the overall performance is to this coefficient.

Please also provide a disclaimer on Tab. 3 that many of the baselines there were apparently not fine-tuned on the APPS data the way RLTF was. The ablations in Tab. 4 show that the performance of RLTF is only modestly better than what may be accomplished by naively fine-tuning CodeT5 on task-specific samples ("Offline"). At a first read, Tab. 3 gives the impression that the model is far better than (much) larger baselines, so this difference in treatment should be emphasized in the table.

Minor comments:
- P1: "up to 70% of the code generated by these models can be non-compilable (Chen et al., 2019)" is not an appropriate citation given that "these models" refers to models introduced at least two years later. The compilation rate is naturally much higher for LLMs than for pre-2020 models. This paper records a rate around 30% in Fig. 2 with a model with just 770M parameters.
- P4, before Eq. 1: superfluous period before colon in "parameters.:"
- Eq. 1: the notation here is quite odd. Please at least fix `max` to use the non-italic form (LaTeX: \max) and add the subscript to indicate that it maximizes over the model's parameters.
- P5: "manifests itself in this category" -- unclear, both the phrasing and the implication that failing unit tests might also not be in this category.  As far as I can tell from the evaluation, this category is exclusively reserved for code with failing tests.
- Tab. 2 uses U_whole but the text on P6 uses U_global. I assume these are the same; please consolidate them.
- Eq. 4: this equation is not discussed by the text, nor is the S/E notation (also used in Eq. 3). I find this notation to be rather clunky in general and would prefer a simpler solution (at the very least for L_coarse, which also sums from start to end). Please at least briefly introduce the interpretation of this equation in the text.
- Eq. 7 - 10: normally the cross-entropy loss is averaged over tokens. Was that not done here? That decision interacts with the choice to add an $\alpha$ term to Eq. 9, which I worry can have an outsized influence on the overall loss when applied to very small snippets.
- Sec. 4.4: this section needs more information on where the GPT results originate from and/or how they were fine-tuned.
- Fig. 2: is quite hard to read. The font is small and the colors are very light. Please increase the font size and contrast.

**Strengths And Weaknesses:**

Strengths:

+ The results are ablated and analyzed in substantial detail, which helps the work provide fine-grained insight into the role and impact of loss terms that operate at different semantic levels (e.g., learning from compilation errors, test failures, both binary and in terms pass rate) on code generation performance.
+ The setup is relatively conventional and easy to follow, allowing for reasonable comparisons with recently published work on RL for code generation.

Weaknesses:

- The work includes a wide array of seemingly hand-picked and unablated constants. Given the narrow scope of the evaluation in terms of datasets assessed (the main results all focus on APPS and MBPP), this raises concerns around overfitting to the evaluation set.

---

> ### Author Response · Authors · 2023-08-21
>
> Thank you for your constructive comments. We sincerely appreciate your time spent reading this paper, and we provide a point-by-point response to your comments, as follows.
>
> Q1: Tab. 2/P6: involves a manual categorization of error types for just a subset of the errors in the Python programming language. This both means that porting this approach to new languages will be time-consuming and that it may not yield the observed improvements in systems with other error types. While those may be rare in interview problems, they abound in practice. The work should discuss this and note it as a limitation. Furthermore, the reason for having a separate U_ignore from U_global/U_whole is unconvincing. The motivation for this category is that "the reasons for the error are numerous and cannot be accurately located". That sounds very much like the definition of a global error. Please provide results where this issue type is grouped under global.
>
> A1: Thank you for raising the issue of transferability. We agree that it is a crucial aspect to consider in our research. Manual categorization of sub-error types make it challenging to transfer RLTF to other programming languages.  The implementation of automated categorization will significantly enhance the applicability of RLTF in a broader context. We have discussed this explicitly in the "Conclusions and Future Work" section.
>
> The code in $U_{global}$ contains mainly logic errors that are not reported by the compiler with specific location feedback. For instance, recursion errors are reported with a general message 'RecursionError: maximum recursion depth exceeded'. On the other hand, the code in $U_{ignore}$ has a feedback of error locations, but these locations do not necessarily correspond to the source of the error. As an analogy, consider a code segment with 20 lines that contains a single triple quote on somewhere. If we run the code, the compiler will report an 'unterminated triple-quoted string literal (detected at line 20)'. However, the actual cause may be that there is an extra or a missing triple quote somewhere in line 1-20. We cannot obtain the exact location from compiler feedback. This is why it "cannot be accurately located", so we choose to ignore them in fine-grained feedback. In the experiments, we observed that the proportion of generated code belonging to $U_{ignore}$ is less than 2% (see 'else' in Fig.2b), and did not see obvious indicator changes when it was grouped under $U_{global}$.
>
> Q2: The work uses 0.3 as a constant in most of its R_* loss terms. Please provide both (a) an analysis where each occurrence of this constant (and others in Eq. 3) is instead tuned on a validation set (a small grid search over permutations suffices) and the optimal values are applied to the test set, as well as (b) an ablation of each coefficient separately other than R_fine in the style of Tab. 6. The goal is to show how sensitive the overall performance is to this coefficient.
>
> A2: For coarse-grained feedback, we set the value of $R$ to one of four pre-determined values (-1.0, -0.6, -0.3, 1.0) based on the type of error, without additional parameter tuning. This approach aligns with the settings used in prior studies such as CodeRL and PPOCoder, enabling more equitable comparisons. While for fine-grained feedback, we conducted experiments to determine best $R$. Specifically, we found that setting $R$ to values between -0.4 and -0.2 resulted in the best model performance on the validation set (as reported in Tab. 6), and thus selected $R=-0.3$ for subsequent experiments. In the case of adaptive feedback, we aimed to strike a balance between failure and pass with a adaptive $R$ value. To achieve this, we set $R$ as a linear function ranging from -0.3 to 1.0, based on the pass rate of the unit tests. Overall, we can only adjust $R$ in fine-grained feedback, as detailed in Tab. 6.

---

> ### Author Response · Authors · 2023-08-21
>
> Q3: Please also provide a disclaimer on Tab. 3 that many of the baselines there were apparently not fine-tuned on the APPS data the way RLTF was. The ablations in Tab. 4 show that the performance of RLTF is only modestly better than what may be accomplished by naively fine-tuning CodeT5 on task-specific samples ("Offline"). At a first read, Tab. 3 gives the impression that the model is far better than (much) larger baselines, so this difference in treatment should be emphasized in the table.
>
> A3: Thanks for your suggestions. In Tab.3, the results of pretrained LMs (except for Codex and GPT3) are from models fine-tuned on APPS using the standard loss $L_{sl}$ only, and results of CodeRL, PPOCoder, RLTF are from models fine-tuned on APPS using the both loss $L_{sl}$ and $L_{coarse}$.  Additionally, we would like to clarify that the results presented in Table 4 ('offline' row) are not obtained through naive fine-tuning. Instead, these results also incorporate the  $L_{sl}$ and $L_{coarse}$  losses, which means that they have undergone coarse-grained reinforcement learning training. We have added this to Section 4.2.
>
> Q4: normally the cross-entropy loss is averaged over tokens. Was that not done here? That decision interacts with the choice to add an \alpha term to Eq. 9, which I worry can have an outsized influence on the overall loss when applied to very small snippets.
>
> A4: Yes, cross-entropy loss in our paper are averaged over tokens. The value of $\alpha$ serves as an adaptive term that balances different reinforcement learning loss terms and is unlikely to have an undue influence. For the sake of simplicity, assume that $bs = 1$, length of generated tokens is $T$, $p(w_{t}|D, \theta, w_{1:t-1})=1$ and $R(W_{baseline})=0$. Under these conditions, the averaged $L_{coarse}$ and $L_{adaptive}$ of each token are equal to $\frac{1}{T} R T = R$, while $L_{fine}$ is given by $\frac{1}{T} \alpha R (E_{fine} - S_{fine})$. To ensure comparability among different loss terms, we set $R=\frac{1}{T} \alpha R (E_{fine} - S_{fine})$, which yields $\alpha=\frac{T}{E_{fine} - S_{fine}}$.
>
> Q5: Other minor issues:
>
> P1: "up to 70% of the code generated by these models can be non-compilable (Chen et al., 2019)" is not an appropriate citation given that "these models" refers to models introduced at least two years later. The compilation rate is naturally much higher for LLMs than for pre-2020 models. This paper records a rate around 30% in Fig. 2 with a model with just 770M parameters.
>
> P4, before Eq. 1: superfluous period before colon in "parameters.:"
>
> Eq. 1: the notation here is quite odd. Please at least fix max to use the non-italic form (LaTeX: \max) and add the subscript to indicate that it maximizes over the model's parameters.
>
> P5: "manifests itself in this category" -- unclear, both the phrasing and the implication that failing unit tests might also not be in this category. As far as I can tell from the evaluation, this category is exclusively reserved for code with failing tests.
>
> Tab. 2 uses U_whole but the text on P6 uses U_global. I assume these are the same; please consolidate them.
>
> Eq. 4: this equation is not discussed by the text, nor is the S/E notation (also used in Eq. 3). I find this notation to be rather clunky in general and would prefer a simpler solution (at the very least for L_coarse, which also sums from start to end). Please at least briefly introduce the interpretation of this equation in the text.
>
> Sec. 4.4: this section needs more information on where the GPT results originate from and/or how they were fine-tuned.
>
> Fig. 2: is quite hard to read. The font is small and the colors are very light. Please increase the font size and contrast.
>
> A5: We thank reviewer for pointing out these issues and we have revised them in our revision.

---

> > ### Comment · Reviewer_fP7d · 2023-09-18
> >
> > I appreciate your detailed response and corresponding updates to the text. One minor note based on your additions: on page 10, you conclude the discussion of PPOCoder's results with "Even when directly using the original results reported in the original
> > paper, our proposed method still outperforms their approach". Looking at Tab. 3, this is mostly incorrect: PPOCoder(original) outperforms RLTF on all pass@1 tasks and on "Comp" in pass@5. The bold-font in the table does not reflect this. Please update both, or update the former and add an explanation for the choice not to highlight the best results in Tab. 3, under the "CS: w/o" condition, to its caption.

---

> > > ### Author Response · Authors · 2023-09-20
> > >
> > > Thank you for your suggestions. We have updated the corresponding statement in Section 4.2 as follows: "Even without selecting the scaled results, our performance is better than the results reported in the original paper, except for pass@1." To provide readers with a clear understanding of the result sources, we concurrently included the results for PPOCoder(original) and PPOCoder(scale) in Table 3. Here, PPOCoder(original) represents the results from the original paper; however, since we did not directly compare against it, we have shaded it in gray. PPOCoder(scale), on the other hand, denotes the scaled results. In addition, we have deleted the highlight result under the "CS: w/o" condition.We have also added this to Section 4.2.

---

### Review · Reviewer_g3kU · 2023-08-10

**Summary Of Contributions:**

The paper proposes a concrete method for applying reinforcement learning to LLMs pre-trained on code, with a reward function derived from datasets that include natural language program specifications, code which implements those specifications, and unit tests. The method combines supervised fine-tuning using the code provided in the datasets, along with RL from a reward function formed by the use of the unit tests. The authors report that the use of reinforcement learning, and the different components of the reward function, together improve performance on the APPS and MBPP datasets compared to prior work.

**Audience:**

Yes

**Broader Impact Concerns:**

No specific concerns.

**Claims And Evidence:**

No

**Requested Changes:**

I would suggest fixing thse minor points:
- Decoder-only autoregressive LMs don't use "masked language modeling". It is inaccurate to refer to them that way.
- In equation 1, please use `\max` instead of `max` in your formula so that it looks like $\max$ and not $max$.
- In a Markov process, the outcome from a given state depends only on the current state, and not the past history. As such, it seems inaccurate to me to refer to program synthesis with an autoregressive LM as a "Markov process", even if it is technically accurate if you define the entire history of the generation so far as the state.

To addresses the weakness above, I recommend the following changes:
- Better describe how the comparative methods' results for Table 3 were obtained to explain why they don't match the numbers from the prior papers.
- Describe the work using more standard RL terminology and either further explain what makes this work "online" and CodeRL/PPOCoder "offline", or remove that part of the paper.
- Describe the size of the online buffer, when things are added to it, and when things are removed from it.
- Describe further how Critic Sampling was used/implemented for this work.

**Strengths And Weaknesses:**

# Strengths
The paper achieves empirical results which are reported to exceed that of the prior work. The proposed method is relatively simple to implement, as it largely amounts to using a particular design of the reward function on top of the policy gradient method.

# Weaknesses
Table 3's results for the prior work don't match what was reported in those papers. For example, I thought that the [CodeRL](https://arxiv.org/pdf/2207.01780v3.pdf) paper's Table 1 (page 12)'s row for "CodeRL+CodeT5 770M" should match Table 3's "CodeRL 770M w/ CS" row, but the numbers are different. Similarly, the numbers for the [PPOCoder](https://arxiv.org/pdf/2301.13816v4.pdf) paper's Table 3 (page 11)'s row for "PPOCoder+CodeT5 700M" don't match the ones in Table 3. The reason for the discrepancy is unclear.

The paper doesn't use terminology/framing commonly used in other reinforcement learning work, which makes it harder to put into context with the prior work. Examples:
- The use of "online" and "offline" seem inconsistent in the paper. In other work, I believe that "online" generally corresponds to using an "on-policy" method and "offline" generally corresponds to using an "off-policy" method. Indeed, this paper uses policy gradient/REINFORCE, which is an on-policy RL method. But CodeRL and PPOCoder also use on-policy RL methods and they are marked as "offline". It wasn't clear to me what makes this method "online" and the others "offline".
- The paper doesn't mention that Equation 2 corresponds to policy gradient/REINFORCE or include any citations to that effect.

The use of $S$ and $E$ in don't seem to make theoretical sense, when $U_{line}$ is used and therefore $S$ and $E$ only covers a small portion of the generated program. For example, if a name error occurs on line 10, it could be because the code on line 10 used some variable or function which wasn't defined earlier in lines 1-9. One way to think of this error is that line 10 should be fixed to only refer previously defined variables. But another way is that lines 1-9 could be revised to have defined the name needed on line 10 (for example, by importing that function from a library).

The description of the "online buffer" is quite sparse. Section 3.2 doesn't describe the size of the buffer, when things are added to the buffer, and when things are removed.

Table 1 says that the proposed method employs Critic Sampling, but there appears to be no description later in the paper of how it was incorporated. For example, there is no mention of training a critic model for the programs, as described in Section 3.3.4 of the [CodeRL paper](https://arxiv.org/pdf/2207.01780v3.pdf).

---

> ### Author Response · Authors · 2023-08-21
>
> Thank you for your constructive comments. We sincerely appreciate your time spent reading this paper, and we provide a point-by-point response to your comments as follows.
>
> Q1: Better describe how the comparative methods' results for Table 3 were obtained to explain why they don't match the numbers from the prior papers.
>
> A1:
> Firstly, for the results without Critic Sampling, we used the official open-source CodeRL model and employed nucleus sampling at a temperature of 0.6. We observed that the pass@1 was lower than what was reported in their paper, while the pass@5 was higher. Similar results were also found by others and discussed at "https://github.com/salesforce/CodeRL/issues/30". Regarding PPOCoder, the results reported in their paper also differ from those in the CodeRL paper. Since the authors did not open-source the APPS benchmark code and models, we are uncertain about their exact evaluation process. Therefore, we used the results obtained from the open-source CodeRL model as a standard and employed proportional scaling to derive the final results. This ensures a fair comparison between the different methods, while acknowledging the discrepancies in the reported values. Even without selecting the scaled results, our performance is better than the results reported in the original paper, except for pass@1.
> | Method              | Size | pass@1 - Intro | pass@1 - Inter | pass@1 - Comp | pass@1 - all | pass@5 - Intro | pass@5 - Inter | pass@5 - Comp | pass@5 - all | pass@1000 - Intro | pass@1000 - Inter | pass@1000 - Comp | pass@1000 - all |
> |---------------------|------|----------------|----------------|---------------|--------------|----------------|----------------|---------------|--------------|-------------------|-------------------|-----------------|----------------|
> | CodeRL              | 770M | 4.00           | 0.78           | 0.15          | 1.30         | 9.83           | 2.03           | 0.69          | 3.32         | 35.30             | 13.33             | 13.60           | 17.78          |
> | PPOCoder(original)  | 770M | 5.20           | 1.00           | 0.50          | 1.74         | 9.10           | 2.50           | 1.20          | 3.56         | 35.20             | 13.35             | 13.90           | 17.77          |
> | PPOCoder(scale)     | 770M | 4.06           | 0.79           | 0.15          | 1.32         | 9.97           | 2.06           | 0.70          | 3.37         | 35.42             | 13.37             | 13.65           | 17.84          |
> | RLTF                | 770M | 4.16           | 0.97           | 0.20          | 1.45         | 10.12          | 2.65           | 0.82          | 3.78         | 38.30             | 15.13             | 15.90           | 19.92          |
>
> Secondly, for the results obtained after Critic Sampling, please refer to A4. We have added this to Section 4.2.
>
> Q2: Describe the work using more standard RL terminology and either further explain what makes this work "online" and CodeRL/PPOCoder "offline", or remove that part of the paper.
>
> A2: In this context, "online/offline" refers to whether the agent can interact with the environment and generate data in real-time. "On-policy/off-policy" concerns whether the algorithm employs data from the current policy to train the model. In our paper, RLTF allows interaction with the environment, enabling online data generation. In contrast, CodeRL and PPOCoder utilize offline training data, prohibiting direct interaction with the environment.
>
> Q3: Describe the size of the online buffer, when things are added to it, and when things are removed from it.
>
> A3: We conducted our training using 8 * V100 GPUs, with a batch size of 32 per GPU. We updated the online buffer every 50 steps. Following the same approach as CodeRL, half of the steps were for SL training, while the other half focused on RL training. Consequently, we set the size of the online buffer to $(50/2) * 32 * 8 = 6400$, and the online buffer was updated as a queue. After generating 6400 samples using the initial model, the training process officially commenced. We have added this to Section 4.1.

---

> > ### Comment · Reviewer_g3kU · 2023-09-18
> > **Response to comment**
> >
> > Thank you for taking the time to answer the questions. I was wondering about the following:
> >
> > > Since the [PPOCoder] authors did not open-source the APPS benchmark code and models, we are uncertain about their exact evaluation process. Therefore, we used the results obtained from the open-source CodeRL model as a standard and employed proportional scaling to derive the final results. This ensures a fair comparison between the different methods, while acknowledging the discrepancies in the reported values.
> >
> > Could you explain in greater detail what you mean by "employed proportional scaling to derive the final results"? It appears to me that the "PPOCoder(scale)" row has the same values as the "CodeRL" row but multiplied by a small constant close to 1.015. If that is accurate, how was this constant chosen?

---

> > ### Comment · Reviewer_g3kU · 2023-09-18
> > **Response to comment**
> >
> > > In our paper, RLTF allows interaction with the environment, enabling online data generation. In contrast, CodeRL and PPOCoder utilize offline training data, prohibiting direct interaction with the environment.
> >
> > My understanding is that CodeRL and PPOCoder also feature direct interaction with the environment.
> >
> > ## CodeRL
> > [CodeRL's paper](https://arxiv.org/pdf/2207.01780v3.pdf) states in Section 3.4 (emphasis mine):
> > > we applied imitation learning to first warm-start a pretrained LM model with $L_{ce}$ only for up to 10 epochs. We then sampled program sequences from this actor network to train the critic while keeping the parameters of the actor network frozen. [...] , we also use the ground-truth programs of training samples to train the critic. These samples are considered perfect programs and always have a label of PassedTest. **After training the critic, we then apply both $L_{ce}$ and $L_{rl}$ with equal weights to finetune the actor network. [...]  in each training optimization step, we can simply approximate the expected gradient with a single sample $W_s \sim p_\theta$:**
> >
> > I believe the last part implies that $L_{rl}$ is computed by 1) sampling one program from the actor $p_\theta$, with the current value of $\theta$ and 2) computing the loss, with the reward function, by executing the sampled program. My understanding from this description is that CodeRL's procedure is much like having the online buffer proposed in this work, but with a buffer size of 1, and updating the buffer after 1 step.
> >
> > However, when I briefly looked through https://github.com/salesforce/CodeRL, they appeared to generate all the code in one batch and didn't regenerate the code from the actor after each update of the actor's parameters.
> >
> > ## PPOCoder
> > Similarly, Algorithm 1 in [PPOCoder](https://arxiv.org/pdf/2301.13816v4.pdf) has a loop where it 1) samples one program from the current policy $\pi_\theta$ on line 5, 2) executes the program to get various rewards on lines 7-10, and then 3) updates the parameters before the next sampling of a program on line 21; this also seems analogous to having an online buffer with a size of 1 and updating the buffer after 1 step.
> >
> > In PPOCoder's code, I believe [lines 195-197](https://github.com/reddy-lab-code-research/PPOCoder/blob/3d67dccff323c3e97be73f2631495891b2c4c69f/rl_run.py#L195-L197) correspond to step 1, [line 205](https://github.com/reddy-lab-code-research/PPOCoder/blob/3d67dccff323c3e97be73f2631495891b2c4c69f/rl_run.py#L205) corresponds to step 2, and [line 217](https://github.com/reddy-lab-code-research/PPOCoder/blob/3d67dccff323c3e97be73f2631495891b2c4c69f/rl_run.py#L217) corresponds to step 3.
> >
> > ## Summary
> > To my knowledge, CodeRL and PPOCoder both claim that they use what I would consider an "online" RL method where the agent can interact with the environment and generate data in real-time. They are "online" because they both feature a loop where 1) the policy is used to sample 1 program, 2) the sampled program is run against the environment to collect rewards, and 3) the rewards are used to update the policy for the next iteration. I wasn't sure about how CodeRL's code implements the procedure described in the paper, but don't know enough to make a definitive conclusion about CodeRL's code.

---

> > > ### Author Response · Authors · 2023-09-20
> > >
> > > Online: LLM generate training samples and get feedback from environment in real-time during training.
> > >
> > > Offline: LLM generate training samples once and reuse them to update model.
> > >
> > > In accordance with the definition provided, CodeRL is categorized as an offline method due to its reliance solely on data from the "epoch_iterator" ([L1331](https://github.com/salesforce/CodeRL/blob/main/trainers/trainer_rl.py#L1331)), without updates during the training loop ([L1285](https://github.com/salesforce/CodeRL/blob/main/trainers/trainer_rl.py#L1285)).
> > >
> > > Additionally, since our code is developed by improving upon the CodeRL baseline, we encountered an issue with both CodeRL and APPS during the implementation process. This issue is mentioned in Section 4.1 of our paper:
> > >
> > > > Note that, during the implementation of our online framework, we encountered an issue in the APPS open-source code, specifically in the unit test portion. If the generated code raised a "segmentation fault" error, it could not be bypassed using "try" and "except", resulting in the main program getting stuck and disrupting the sample generation process. To address this issue, we modified all unit test code procedures to be executed through subprocess calls, ensuring the main program does not get stuck and allowing the online sample generation process to run smoothly. We believe it necessary to expose this to the community for reproducibility considerations.
> > >
> > > After fixing this issue, we were able to perform online generation, which further confirms that the open-source code of CodeRL is offline.
> > >
> > > Conversely, PPOCoder's open-source code suggests it is an online method. We acknowledge and apolopy for our previous error, and have rectified the erroneous description from our paper. Moreover, the open-source code of PPOCoder is designed for code translation tasks, rather than for benchmarks like APPS that include unit tests. This can be inferred from the [reward definition](https://github.com/reddy-lab-code-research/PPOCoder/blob/main/reward.py#L125)(Only +1 and -1 are used, as shown in Eq. 8 in the paper, rather than Eq. 7) and other aspects, as well as the absence of any evaluation code related to APPS. Secondly, their paper also does not mention the issue we discussed above, which led us to mistakenly assume that their approach for the APPS benchmark was an offline method. Furthermore, our experiments in Table 4 of the paper show that, online methods generally outperformed offline ones. However, the results in Table 3 of the PPOCoder paper indicate that the combined improvements from all their modifications (including online, PPO algorithm, DFG, AST, etc.) are only slightly better than those of CodeRL (offline). This is another reason why we initially mistakenly assumed that their method was an offline approach.
> > >
> > > Within our proposed RL framework, online training is a crucial component that, when combined with multi-granularity feedback (Tab. 4), enables distinct LLM models (such as CodeT5 and CodeGen) to achieve varying levels of performance enhancement.

---

> > ### Author Response · Authors · 2023-09-20
> >
> > Thank you for your comments. CodeRL has open-sourced their model, and we have evaluated it to obtain the results for CodeRL(opensource), corresponding to the first row in the table below. Since PPOCoder has not released their model, we have referred to the results from the original [PPOCoder paper](https://arxiv.org/pdf/2301.13816.pdf), page 11, Table 3: CodeRL(paper) and PPOCoder(paper), which correspond to the second and third rows in the table, respectively. Finally, We obtain the final result PPOCoder(scale) using the formula: $PPOCoder_{scale}=  CodeRL_{opensource} * \frac{PPOCoder_{paper}}{CodeRL_{paper}}$, corresponding to the fourth row in the table. As the results from the PPOCoder's original paper are approximately 1.015 times those of CodeRL, you may notice that the "PPOCoder(scale) row has the same values as the "CodeRL" row but multiplied by a small constant close to 1.015".
> > | Method              | Size | pass@1 - Intro | pass@1 - Inter | pass@1 - Comp | pass@1 - all | pass@5 - Intro | pass@5 - Inter | pass@5 - Comp | pass@5 - all | pass@1000 - Intro | pass@1000 - Inter | pass@1000 - Comp | pass@1000 - all |
> > |---------------------|------|----------------|----------------|---------------|--------------|----------------|----------------|---------------|--------------|-------------------|-------------------|-----------------|----------------|
> > | CodeRL(opensource)              | 770M | 4.00           | 0.78           | 0.15          | 1.30         | 9.83           | 2.03           | 0.69          | 3.32         | 35.30             | 13.33             | 13.60           | 17.78          |
> > | CodeRL(paper)              | 770M | 4.90           | 1.06           | 0.5          | 1.71         | 8.60           | 2.64           | 1.0          | 3.51         | 36.10             | 12.65             | 13.48           | 17.50          |
> > | PPOCoder(paper)  | 770M | 5.20           | 1.00           | 0.50          | 1.74         | 9.10           | 2.50           | 1.20          | 3.56         | 35.20             | 13.35             | 13.90           | 17.77          |
> > | PPOCoder(scale)     | 770M | 4.06           | 0.79           | 0.15          | 1.32         | 9.97           | 2.06           | 0.70          | 3.37         | 35.42             | 13.37             | 13.65           | 17.84          |
> > | RLTF                | 770M | 4.16           | 0.97           | 0.20          | 1.45         | 10.12          | 2.65           | 0.82          | 3.78         | 38.30             | 15.13             | 15.90           | 19.92          |

---

> ### Author Response · Authors · 2023-08-21
>
> Q4: Describe further how Critic Sampling was used/implemented for this work.
>
> A4: Due to the unavailability of the open-source Critic Sampling code from CodeRL, we reproduced the Critic Sampling based on the details provided in the original paper. We employed both refine and repair and set $M$, the number of top candidates selected from program samples that fail example unit tests to $1$. We also limited the maximum number of repair and refine iterations to $1$. It is worth noting that we found a crucial parameter $N$ not explicitly mentioned in the original paper: the total number of program samples generated during each refine/repair step. When measuring $pass@1$, we set $N=5$; for $pass@5$, we set $N=20$; and for $pass@1000$, we set $N=1000$. Our results show that Critic Sampling achieves better performance when $k=1$ or $5$ in $pass@k$. However, when $k$ is larger, the improvement in performance is relatively small, which differs from the results presented in the original paper. Overall, we think Critic Sampling is a post-processing method that focuses on how to better use the model rather than how to better train the model. Therefore, comparing our approach with results without Critic Sampling provides a fairer evaluation. We have added this to Section 4.2.
> Finally, for the results obtained after Critic Sampling, we have used the results from the [CodeRL](https://proceedings.neurips.cc/paper_files/paper/2022/file/8636419dea1aa9fbd25fc4248e702da4-Paper-Conference.pdf) published at NeurIPS 2022, page 8, Table 1(a) before. We have now updated Table 3 to include the latest results you provided.
>
> Q5: The use of $S$ and $E$ in don't seem to make theoretical sense, when $U_{line}$ is used and therefore $S$ and $E$ only covers a small portion of the generated program. For example, if a name error occurs on line 10, it could be because the code on line 10 used some variable or function which wasn't defined earlier in lines 1-9. One way to think of this error is that line 10 should be fixed to only refer previously defined variables. But another way is that lines 1-9 could be revised to have defined the name needed on line 10 (for example, by importing that function from a library).
>
> A5: The purpose of the $S$ and $E$ is to provide precise feedback to assist the LLM in identifying the location of the error, allowing them to focus solely on fixing the error. However, as demonstrated in the example provided, there are multiple approaches to correcting errors. We do not advocate for a specific method but rather consider any approach acceptable as long as the unit test is passed successfully. Once corrected, the penalty associated with the error will be resolved from the LLM's perspective. Consequently, the LLM will continue to explore possible solutions until the error is resolved.
>
> Q6: Other minor issues:
>
> ●Decoder-only autoregressive LMs don't use "masked language modeling". It is inaccurate to refer to them that way.
>
> ●In equation 1, please use \max instead of max in your formula so that it looks like \max and not $max$.
>
> ●In a Markov process, the outcome from a given state depends only on the current state, and not the past history. As such, it seems inaccurate to me to refer to program synthesis with an autoregressive LM as a "Markov process", even if it is technically accurate if you define the entire history of the generation so far as the state.
>
> A6: Thanks for your suggestions. We have revised these issues in our revision.

---

### Review · Reviewer_fu8D · 2023-08-22

**Summary Of Contributions:**

This paper proposes RLTF, a reinforcement learning framework for training code LLMs using fine-grained unit test feedback. Specifically, the reward includes 3 terms: (1) coarse-grained feedback that assigns a positive value for correct code, and different negative values for different errors, same as the CodeRL work. The reward is equally assigned to all code tokens; (2) fine-grained feedback. This feedback comes from the runtime error or compiler error that identify the locations of the errors, thus the negative reward is only applied to the code line that contains errors; and (3) adaptive feedback. This reward term depends on the percentage of passed unit tests. They perform RLTF on both CodeT5 and CodeGen, and evaluate their approaches on APPS and MBPP. They demonstrate that RLTF consistently improves the performance, and outperforms CodeRL and PPOCoder.

**Audience:**

Yes

**Broader Impact Concerns:**

No concern.

**Claims And Evidence:**

No

**Requested Changes:**

1. Please clarify the differences between RLTF and prior work in terms of the RL design. To my knowledge, CodeRL and PPOCoder also utilizes the LLM to generate code for computing rewards during training. Therefore, I think the main difference between RLTF and previous work is the reward design, not from the online/offline aspect.

2. Please add the ablation result with all feedback except the coarse one.

3. Why RLTF increases the failure? Does it mean that RLTF mainly decreases the runtime errors and compiler errors, but does not help reduce the semantic errors?

4. Please also present pass@1 results on MBPP.

**Strengths And Weaknesses:**

Strengths

1. I think the overall idea of this work is promising and useful for improving code generation. Unit test execution provides rich fine-grained information, and leveraging such feedback with reinforcement learning is beneficial.

2. Results on both APPS and MBPP are good and outperform prior RL approaches for training code LLMs.

3. The ablation studies are informative and demonstrate the importance of different design choices, such as different feedback types. The error analysis also demonstrates the improvement of RLTF on problems of different difficulty levels, and the distribution of different error types.

Weaknesses

1. First, I would like the authors to clarify the differences between RLTF and prior work in terms of the RL design. To my knowledge, CodeRL and PPOCoder also utilizes the LLM to generate code for computing rewards during training. Therefore, I think the main difference between RLTF and previous work is the reward design, not from the online/offline aspect.

2. The ablation on different feedback combinations is helpful and shows that leveraging finer-grained execution feedback improves the performance. However, I feel that the coarse feedback becomes redundant after adding fine-grained and adaptive feedback. Could the authors try the ablation with all except the coarse feedback?

3. From the error analysis, RLTF seems to increase the failure. What is the explanation for this? Does it mean that RLTF mainly decreases the runtime errors and compiler errors, but does not help reduce the semantic errors?

4. On MBPP, why only pass@80 was reported? It is helpful to also present pass@1 results.

---

> ### Author Response · Authors · 2023-08-31
>
> Thank you for your constructive comments. We sincerely appreciate your time spent reading this paper, and we provide a point-by-point response to your comments as follows.
>
> Q1: Please clarify the differences between RLTF and prior work in terms of the RL design. To my knowledge, CodeRL and PPOCoder also utilizes the LLM to generate code for computing rewards during training. Therefore, I think the main difference between RLTF and previous work is the reward design, not from the online/offline aspect.
>
> A1: Similar to CodeRL, we employ policy gradient as the fundamental RL algorithm in our approach. However, our method differs from CodeRL and PPOCoder in that the training process involves real-time interaction between the LLM (agent) and the environment, and generates data online. In contrast, CodeRL and PPOCoder rely on offline training data and do not interact with the environment, as evidenced in their open-source code([CodeRL](https://github.com/salesforce/CodeRL), [PPOCoder](https://github.com/reddy-lab-code-research/PPOCoder)). Moreover, the most significant distinction between RLTF and previous work lies in the reward design, which we have already highlighted in the introduction section of our paper.
>
> Q2: Please add the ablation result with all feedback except the coarse one.
>
> A2: The ablation results with all feedback except the coarse one are as follows: pass@1: 1.38%, pass@5: 3.58%, pass@10: 5.05%. The purpose of the coarse feedback is to inform the model about the overall test results of the code (true, false, runtime error, compile error). Therefore, we think it is also quite important. As can be seen from the results, removing the coarse feedback leads to a decline in the performance metrics. It is also worth noting that the performance without coarse feedback is better than the performance with only coarse feedback. We will add these results into our paper.
>
> Q3: Why RLTF increases the failure? Does it mean that RLTF mainly decreases the runtime errors and compiler errors, but does not help reduce the semantic errors?
>
> A3: We would like to clarify that the observed increase in failure rate stems from the fixing of error codes, resulting in either pass or fail outcomes. Results in Fig.2 illustrate that our approach is more effective in addressing runtime and compiler errors compared to semantic errors, which remain more challenging. We have developed fine-grained feedback for runtime and compiler errors and adaptive feedback for semantic errors, which, when combined, have resulted in improved model performance. Through ablation experiments presented in Tab. 5, we have demonstrated RLTF's varying degrees of success in addressing all three types of errors.  We agree that providing semantic error information directly to LLM is a crucial research direction, and we will continue to explore ways to enhance LLM's capacity to understand and address semantic errors. We will include these discussions into our paper.
>
> Q4: Please also present pass@1 results on MBPP.
>
> A4: We obtained pass@1 results on MBPP at a temperature of 0.1, which measured 30.35%. Using the open-source CodeRL model at the same temperature, we obtained a pass@1 result of 25.68%. The pass@1 result reported in the PPOCoder paper（https://openreview.net/forum?id=0XBuaxqEcG） is 26.1%. As can be seen, the performance of our model in the pass@1 metric is significantly higher than that of previous work, demonstrating the effectiveness of our proposed method.

---

### Author Response · Authors · 2023-10-18

Dear reviewers,

We'd like to thank all of you for the constructive feedback, and we had revised our paper accordingly by Sept. 20th.
Should you have any further concerns, please let us know.

Best

Paper authors

---

### Decision · Action_Editor_TY24 · 2023-10-27

**Recommendation:** Accept with minor revision

**Comment:**

Several recent papers have used RL to finetune LLMs for code, and the paper makes a useful contribution to this space. Specifically, the reviewers liked the reward design ideas introduced in the paper and found the empirical results to be impressive. However, there were concerns about the claim that this is the first "online" RL approach to code model tuning. The authors have since admitted that PPOCoder is also an online method, and the reviewers were happy with their response. However, the early parts of the paper don't reflect this fact appropriately yet, and the paper needs another deep pass over these sections. Given this, I am recommending acceptance with minor revisions.

**Audience:**

The paper makes interesting technical contributions to ML for code, an increasingly important subarea of machine learning. Given this, it is likely to interest a nontrivial subset of TMLR's audience.

**Claims And Evidence:**

The paper offers a new framework (RLTF) that uses RL feedback from execution on unit tests to refine code LLMs. While the idea of using RL to tune code LLMs is not new at this point, the present paper claims to be different on two counts: (1) it uses online, rather than offline, RL and (2) its reward function is finer-grained than in previous work --- in particular, it specifically penalizes the erroneous parts of generated programs. The main claim is that this methodology leads to better performance in code generation.

The claims about the method's overall benefits are convincing. Specifically, the paper offers comparisons against multiple prior RL-based methods, using the same benchmarks (APPS and MBPP) as in those prior works. The results indicate state-of-the-art performance in code generation.  There were concerns about the paper's claim of being the first "online" RL approach in this setting. However, these concerns were mostly addressed during the discussion.